# Multi-Parameter Predictive Model of Mobile Robot’s Battery Discharge for Intelligent Mission Planning in Multi-Robot Systems

**DOI:** 10.3390/s22249861

**Published:** 2022-12-15

**Authors:** Bartosz Poskart, Grzegorz Iskierka, Kamil Krot, Robert Burduk, Paweł Gwizdal, Arkadiusz Gola

**Affiliations:** 1Faculty of Mechanical Engineering, Wrocław University of Science and Technology, ul. Łukasiewicza 5, 50-371 Wrocław, Poland; 2Faculty of Information and Communication Technology, Wrocław University of Science and Technology, ul. Z. Janiszewskiego 7, 50-372 Wrocław, Poland; 3Faculty of Mechanical Engineering, Lublin University of Technology, ul. Nadbystrzycka 36, 20-618 Lublin, Poland

**Keywords:** autonomous mobile robots, battery consumption, predictive mission assignment, logistics 4.0, predictive monitoring, AGV, AMR

## Abstract

The commercially available battery management and mission scheduling systems for fleets of autonomous mobile robots use different algorithms to calculate the current state of charge of the robot’s battery. This information alone cannot be used to predict whether it will be possible for a single robot in the fleet to execute all of the scheduled missions. This paper provides insight into how to develop a universal battery discharge model based on key mission parameters, which allows for predicting the battery usage over the course of the scheduled missions and can, in turn, be used to determine which missions to delegate to other robots in the fleet, or if more robots are needed in the fleet to accomplish the production plan. The resulting model is, therefore, necessary for mission scheduling in a flexible production system, including autonomous mobile robot transportation networks.

## 1. Introduction

The autonomous logistics solutions available on the market today, like the ones offered by Mobile Industrial Robots, AGILOX, Omron, KUKA and many others, rely on the use of real-time data for mission scheduling [1,2]. The main problem with such solutions is the management of a fleet of transport robots based purely on simple scheduling of missions and charging only when its battery level drops below a strictly defined threshold, which may disrupt the supply chain of the production system [3].

The commonly used Autonomous Mobile Robots (AMRs) allow for rapid reprogramming of transportation tasks and have the ability to flexibly change their route depending on the conditions and obstacles in the production hall [4]. Such autonomy provides many positive aspects but also causes some problems due to the inability to fully predict real routes and times of transportation [5]. Power management systems for such fleets of mobile robots are currently based on simplified models and do not take into account all of the parameters that may affect the battery discharge characteristic [6]. To fully exploit the potential of AMRs and optimise transport, an AMR intelligent fleet management system is needed that can not only be integrated with manufacturing Information Technology (IT) and Operational Technology (OT) systems but also provides oversight of production operations based on real-time information [7]. A crucial element of such a multi-robot system is an iterative predictive algorithm verifying the mission execution feasibility based on the production process data [8].

The problem, therefore, comes down to developing a model estimating the battery charge consumption during the operation of a mobile robot, depending on the type of the operation performed, which will allow for the appropriate action to be taken in advance like scheduling the robot’s charging or delegating missions to other robots in the fleet. The purpose of this paper is to determine the factors that have a significant impact on the battery consumption of the MiR100 mobile robot, to develop a predictive model for the battery discharge based on the assigned missions, and to compare the models developed this way for two different batteries from the same manufacturer that differ in the production date and the level of exploitation.

Many of the first automated intralogistics systems were based on Automated Guided Vehicles (AGVs), where the distribution of tasks between individual units and the planning of their charging schedule are closely related to the topography of the environment and their navigation method. AGV-based solutions have resulted in the development of logistics management methods designed around these technical limitations. Currently, logistics systems based on AGVs are being replaced by the ones using AMRs instead. The main advantage of such systems is no requirement to build a dedicated infrastructure for their agents, which is required in the case of AGV-based systems. AMR navigation is performed with the use of Simultaneous Location and Mapping (SLAM) systems based on laser, vision and acoustic sensors, where no additional reference points, guiding lines or wires are required. The differences between AGV and AMR technologies have been extensively presented in the publication by Fragapane et al. [9].

In-house transportation with the use of AGVs is less flexible in relation to AMRs, because a failure of one of the robots or an obstacle on the AGV’s route may lead to creating congestions, bottlenecks or even cutting the supply chain to the workstations. AMRs use more advanced navigation technologies, and they do not require costly modifications to the plant’s infrastructure (installation of guiding wires, magnetic strips, mechanical and optical tracks), due to which their implementation is much cheaper. There are no costly production disruptions in the implementation of AMRs, and the return on investment (ROI) is usually around six months. All this causes AMRs to replace the AGV technology on the market increasingly.

Due to a completely different approach to navigation of both technologies, the strategies used for AGV management systems are not optimal for AMR fleets, which has contributed to the development of new algorithms for navigation, mission allocation, charging schedule planning, battery management policy, the use of plant infrastructure and so on. Most of these algorithms are based on the parameters of the mobile robots used, such as the current battery charge level, speed, and transported load, which ultimately translate into the consumption of the battery charge. The problem with current AMRs is the use of information from the Battery Management System (BMS) only in the context of the current battery charge consumption and the lack of estimation based on future variable consumption for all of the assigned missions.

One of the areas where the information on the current status of the battery is used is the scheduling of charging autonomous mobile robots within a fleet. The commercially available software offered by robot manufacturers very often allows only setting a lower threshold of SoC, below which the robot is sent back to the charging station, which offers little flexibility of the system and may involve disruption of the supply chain inside the plant or the need to provide a buffer in the form of a replacement mobile robot. For more accurate scheduling, an improved universal battery discharge model should be defined.

Attempts to model the discharge characteristics of the battery were published in many works. In the simplest form, a deterministic linear model can be adopted, as presented by Mei et al. in [10]. In recent years, non-linear battery discharge models have also been used for Unmanned Aerial Vehicles (UAVs) [11] and mobile robots [12]. In previous research works, the authors also attempted to create a non-linear battery discharge model for the executed mission within the entire discharge cycle [13], which is the basis for the research contained in this publication. The main drawback of the previous approach is that the model would need to be defined for each mission separately, and it only involved the SoC analysis for each mission. This publication, on the other hand, focuses on creating a universal model for the battery, including parameters like SoC and the mission and environmental parameters discussed in the following sections.

When creating a battery discharge model, various parameters can influence the amount of battery charge used for specific tasks. In the case presented by Rappaport and Bettstetter [14], the parameter taken into account by the authors is the distance travelled. Often the battery charge level is also estimated using an extended Kalman filter (EKF), where the base parameters are the measured voltage and current of the battery. Such an approach was adopted in publications [15,16]. In another approach to planning missions performed by a mobile robot, the Markov decision process (MDP) is used, where space-time is taken into account, i.e., the topology of the environment that changes with time [17]. Angelo et al. evaluated the performance of several linear models for battery state-of-health estimation and proposed a two-feature model as the best compromise between estimation improvement with respect to single-feature models and collinearity reduction [18].

Another approach to creating a model would be to use multivariable linear regression, which assumes a linear relationship between a set of inputs (features or independent variables) and the outcome (target or a dependent variable). The regression model estimates how the dependent variable changes as the independent variables change. Despite constituting such a simple assumption about the linear relationship between the outcome and a set of inputs, the regression models proved to work well in many different applications, like medicine [19], industrial process measurements [20], aviation material consumption [21] or traffic flow prediction [22]. The multivariable linear regression models have also been used successfully in battery degradation prediction [23,24,25]. The most important advantage of multivariate regression is that it helps understand the correlation between dependent and independent variables. This regression model can also determine the relative influence of one or more predictor variables on the criterion value. The advantage of this model is the ability to identify outliers or anomalies. The multivariate regression models do not have the disadvantages of other machine learning methods, such as overfitting (like Decision Tree Regression, Random Forest Regression), poor results on small datasets (like Decision Tree Regression) or compulsory applying feature scaling (like Support Vector Regression).

In recent years, the provided research was focused on the problem of battery discharge prediction in different applications. Liu et al. proposed both a method for predicting battery SDV-drop based on a pre-classifier [26] and a self-discharge prediction method for lithium-ion batteries based on improved support vector machines [27]. Conte et al. proposed an adaptive method to predict the battery discharge of a multirotor drone over a generic path [28], while Zhao et al. used the evidential reasoning algorithm to fuse the outputs of three typical prediction models to improve the prediction accuracy and verified the proposed method using the NASA battery dataset [29]. Gokcen et al. presented foreseeing of the Lithium-ion battery discharge models for the Internet of Things (IoT) devices under randomised use patterns and also used the NASA Ames prognostics data repository [30]. The research focused on the effects of current rates and ambient temperature on the thermal behaviour of high-energy LiNi0.8Co0.15Al0.05O2//Si-C pouch battery was presented by Zhao et al. [31]. Moreover, Zou et al. proposed an online method based on particle swarm optimisation and support vector regression to estimate the state of health and remaining useful life [32]. Finally, a new method for cycle life and full life cycle capacity prediction was proposed, which combines the early discharge characteristics with the neural Gaussian process (NGP) model proposed by Yin et al. [33].

Unfortunately, the number of publications that focus on the problem of battery discharge in mobile industrial robots is very limited. Therefore, in the following sections, the method of building a model that takes into account important parameters that may affect the discharge characteristics of the Lithium NMC battery of the MiR100 autonomous mobile robot will be presented. The modelling methodology can also be applied to mobile robots from other manufacturers and is not limited to one type of battery. The main difference between the proposed modelling approach presented in this paper and the models presented in the cited works is that they estimate the battery SoC based on the parameters of the battery itself in a given moment under a specified load. The proposed model, on the other hand, estimates the change in the SoC based on the environmental and mission variables.

## 2. Materials and Methods

As a continuation of the work presented by the authors in [13], this paper focuses on measuring the state of charge of two 39.6 Ah Lithium NMC batteries of a MiR100 autonomous mobile robot. While the previous work focused on testing the viability of using a modelled discharge curve for predictive monitoring and mission planning in multi-agent systems with regard to a specific mission, this paper sets out to identify key parameters with the most influence on the battery discharge characteristic. These key parameters, if identified correctly, can be used in a universal function, estimating the battery usage of a specific mission at a specific point in the discharge cycle. In the experiments, the two independent batteries have been tested for redundancy purposes and to test for any inconsistencies between different batches of batteries since one of them was produced in 2018 and the other one in 2020. Figure 1 presents the Ishikawa diagram of potential key parameters.

The parameters taken into consideration were divided into four categories, two of which are mission-dependent, meaning that they may vary depending on the environmental disturbances or the control system variations, and the other two are hardware-dependent, which are related to the kinematics of the robot and its equations of motion, as well as the state of the power source, considering both the short- and long-term changes in the batteries. Not all of the presented parameters can be reliably used in the algorithm due to the specific sensors not being implemented into the used robot or simply because they cannot be justified in a typical industrial environment. Table 1 presents these parameters separated into two groups—the ones that can be measured and justified in a typical industrial environment and those that cannot.

While some of the measurable parameters presented in Table 1, like linear and angular velocity, battery SoC and its time of use, can be acquired directly through the robot’s REST API in the form of a JSON response with the status of the robot. Other parameters, like the distance travelled in a single mission and the number of turns made, can be calculated from the parameters returned by the robot. The payload in a given mission, on the other hand, could be specified for its entire length and controlled in the experiment, though an additional scale could be installed on the robot to measure the weight of the payload throughout the entire mission, should it be variable. While these parameters are given by the robot, the temperature of the battery calculated by the BMS can only be accessed through the web-based interface. In order to take it into account during the experiment, an external temperature sensor was installed inside the robot. To acquire all this data for further analysis, a wireless measuring system had to be built. The system was based on Node-RED, and its architecture is further explained in Section 2.6.

### 2.1. Environmental Key Mission Parameters

In typical industrial applications, the environment in which the mobile robots operate can constantly change due to disturbances caused by human personnel or the changing production and the variable parameters of the mission itself. Moreover, the more robots in the fleet there are, the higher the chance of their interaction or the intersection of their paths, which may lead to a change in key mission parameters and complexity. Due to the autonomous nature of the robot, many of these parameters may be impossible for the programmer to utilise when designing a mission for the robot and will be difficult, or in some cases impossible, to implement in the algorithm due to the closed architecture of the robot’s control system, though some raw parameters can be accessed through the REST API of the mobile robot and others can be calculated from them. This will allow the algorithm to gather, calculate and ultimately use these key parameters to predict the battery usage of the scheduled mission queue. The potential key parameters recognised in this category are the ones that change with the environment and are related to the robot’s trajectory, which ultimately affects its battery life. Those parameters are the number of turns and the distance to travel, which determine the complexity of the default trajectory; the average linear and angular velocity of the robot, which may differ based on some unexpected obstacles or dynamic speed restriction zones; and the payload, which may differ slightly depending on the goods transported during the mission.

### 2.2. Key Mission Parameters

The parameters affected by the control system are very similar to the environmental ones due to the fact that the control system may react to the changing environment. In this case, a smaller variation may be observed due to the slight changes in the input values for the closed control algorithm of the robot. The robot calculates the path each time the mission is called, and the calculations may yield different results each time. Though usually small, there may be some edge cases where the calculated path differs significantly from the default one.

### 2.3. Resistance to Motion

Just like for any other vehicle, an analysis of resistance to vehicle motion can be performed for an autonomous mobile robot. Potential factors to include in the analysis would be the air, gradient and rolling resistances, as well as the inertia of the vehicle. The formula for the resistance to vehicle motion is presented below:(1)$$F_{res}=F_{A}+F_{G}+F_{R}+F_{I},$$
where: *F_res_* is the resistance to motion, *F_A_* is the air resistance, *F_G_* is the gradient resistance, *F_R_* is the rolling resistance, *F_I_* is the inertia.

Upon expanding the above formula, it is possible to determine the basic variable parameters that may affect battery consumption.
(2)$$F_{res}=\frac{C_{D}mgV^{2}}{2A_{F}}+mgsin\alpha \frac{fmg}{RA_{C}}+am,$$
where: *F_res_* is the resistance to motion, *C_D_* is the drag coefficient, *m* is the mass, *g* is the gravitational constant, *V* is the velocity, *A_F_* is the vehicle’s frontal surface area, *α* is the slope angle, *f* is the rolling resistance coefficient, *R* is the wheel radius, *A_C_* is the wheel contact area, *a* is the acceleration.

Analysing the above formula, one may come to the conclusion that parameters like the total mass (changing with the payload), velocity and acceleration may affect the amount of battery consumed in a mission. The magnitude of this effect will be investigated in the results section of this paper.

### 2.4. Power Source

The power source used in the mobile robot may significantly affect the amount of charge used for a specific mission in both the short and the long term. The short-term effects may be caused by the current temperature, the time of continuous use or, as presented in the previous work [13], the current SoC of the battery. The long-term effects may occur due to the state of health (SoH) of the battery due to the level of its exploitation. It is important to note that the type of the battery may also have an effect on its discharge model, but for the purpose of this research, only the Lithium NMC batteries were used since only this type of battery is being used in the MiR mobile robots.

### 2.5. Available Status Parameters

The AMR used in the experiment provides a number of parameters through the REST API, which can be acquired through a specific HTTP request. The parameters available by default are quite simple, and not all of the potential parameters specified before are available. The basic parameters returned in the status HTTP request can be found in the documentation of the REST API of MiR mobile robots [34]. Using the basic parameters provided by the API, like battery percentage, distance travelled, linear and angular velocity, position and orientation, it is possible to calculate most of the parameters needed. The parameters missing from the status response were the weight of the payload and the battery temperature, which needed to be implemented separately in the data acquisition system. The load mass was constant for specific measurement series and was entered manually into the system, while the temperature was measured with an additional temperature sensor installed on the battery.

### 2.6. Measuring System and Communication Protocols

The MiR mobile robot is equipped with a router, which is primarily used to configure and control the robot, as well as manage the mission queue of a single robot (an additional system is necessary to manage a fleet of MiR mobile robots). The wireless connection allows for sending HTTP requests to monitor or control the robot. These requests have been primarily used to gather the basic parameters of the robot, which will be, in turn, used to calculate the more advanced parameters affecting the battery discharge while executing the mission queue. Figure 2 shows a diagram of the prepared measuring system used for data acquisition of individual battery discharge scenarios.

To measure the temperature of the robot’s battery during the discharge cycle, a Raspberry Pi 3B+ microcomputer was installed inside the mobile robot along with the DS18B20 temperature sensor, connected to the GPIO pins of the microcomputer. The AMR and the microcomputer were connected in a local wireless network with an external computer used to manage the queue of missions, acquire process data, process the parameters in the Node-RED platform and store them in the database for future analysis.

The subsystem based on the microcomputer was used to measure the current battery temperature, preprocess the data and send it via an MQTT Mosquitto broker to the main data acquisition computer based on Node-RED. After each mission, the temperature data was averaged and saved to a local MySQL database. The data processing algorithm is presented in the pseudo-code below (Algorithm 1).
**Algorithm 1: Data acquisition and analysis from each mission*****WHILE** true:*   *mission ← getMissionID()*   *currentMission ← mission*   *initialStatus ← getRobotStatus()*   ***WHILE** mission == currentMission:*      *status ← getRobotStatus()*      *avgParams ← countAvgParams(status.linV, status.angV, status.batTemp)*      *currentMission ← getMissionID()*   ***ENDWHILE***   *statusDifference ← calcDifference(status, initialStatus)*   *missionParams ← calcMissionParams(avgParams, statusDifference)*   *DataBase ← (missionParams, avgParams)****ENDWHILE***

The parameters used in creating the mathematical model of a battery are discussed in more detail in the following subsection.

### 2.7. Data Acquisition

In order to investigate the behaviour of the mobile robot’s battery, a series of battery discharge cycles was carried out. For this purpose, three missions of different lengths of the covered route were programmed: 30 m, 140 m and 350 m. Similar travel distances for missions are often encountered in typical industrial applications. For each length, measurements were made with a payload of 0 kg, 50 kg and 100 kg, which is 0%, 50% and 100% of the nominal payload for the MIR100 robot. The numbers of full discharge cycles for both tested batteries with different load configurations and mission lengths are presented in Table 2.

For each variant of the load configuration and mission length, at least two full battery discharge cycles were measured, with the limited number of measurements being dictated by the length of a single discharge and recharge cycle, which takes approximately 16 h. In addition, some of the redundant missions were used in the testing set described in Section 4. It should be noted that the values of some parameters (mission length, average linear speed, average angular speed, number of turns, battery temperature) may differ in the scope of a performed mission due to the changing environment and autonomous routing algorithms of the robot. Figure 3 shows a map of the mobile robot’s working area.

## 3. Results

In accordance with the previous analysis of potential key parameters that affect the battery discharge during a mission, a set of variables was chosen to verify their impact as independent predictors. Table 3 shows the description, label and type of variables. In the conducted research, “Measured energy consumption” was adopted as the dependent variable, while the remaining variables play the role of independent variables, with the exception of the “Date of measurement” variable, which is descriptive only and is not used in the construction of the predictive model.

Below are the research goals set out in this publication:**G1.** Identification of the parameters that have the greatest impact on the characteristics of energy consumption of Lithium NMC batteries used in the MiR100 autonomous mobile robot.**G2.** Finding the optimal model for predicting the consumption of the battery charge level used in the MiR100 autonomous mobile robot based on the key parameters.**G3.** Comparison of the obtained models for two batteries from the same manufacturer that differ in the production date and the total exploitation time.

In order to achieve the set research goals, the collected data and SAS Studio software were used to form a battery discharge prediction model using general multivariate regression (GLM). In the conducted experiments, the number of independent variables is 7 (see Table 3). The mentioned variables are single effects in the analysed GLM model. In the original model, n-way factorisation with n = 2 was used. Additionally, the polynomial order was determined for the value of k = 3. With these assumptions, the tested model consists of 42 effects. The following effects selection algorithms were used to choose the model:Forward selection;Stepwise regression;Least absolute shrinkage and selection operator (LASSO);Least angle regression (LAR).

In the performed experimental studies, the influence of data filtering was also examined concerning the SoC and EnergyCons variables. The adopted data filtering assumptions take into account the following ranges of values of the listed variables:filter 1 (EneryCons > 0.2);filter 2 (SoC < 80 and SoC > 20);filter 3 (SoC < 80 and SoC > 20 and EnergyCons > 0.2).

The use of filter 1 is a result of the rounding of the calculations of the internal BMS, which could not be modified in any way by the authors. This rounding for very short missions with very little energy consumption may lead to misinterpreting their actual energy consumption. In addition, it was noticed that the battery discharge characteristic for missions consuming less than 0.2% of the maximum battery capacity does not fit into the 9th-degree polynomial formulated by the authors in [13] and sometimes appears more random, which is why such missions were not analysed.

The use of filter 2 results from the recommendation of robot manufacturers and the research conducted by the scientific community on Lithium-ion batteries [35,36,37,38], where it has been concluded that optimally the SoC of the battery should be in the range of 20–90%. Additionally, it is worth mentioning that the charging time above 80% of SoC takes longer, so the filter applied to the dataset is limiting the SoC data to the range of 20 to 80%.

While filter 1 limits the data points to missions in which the energy consumption (EnergyCons) was greater than 0.2% of the maximum battery capacity, filter 2 selects the missions by the SoC values in the range of 20–80%. Filter 3 is a logical product of filters 1 and 2.

Table 4 presents the results for each of the considered effects selection algorithms for the proposed data filters and both of the tested batteries. The Adjusted R-Square (AR-S) index was adopted as the model selection criterion. The obtained results show the effectiveness of the GLM method in the problem of predicting battery consumption. The values of the AR-S indicator are close to the maximum value of 1, which proves a very good fit for the model, i.e., the correct prediction of battery consumption. The best results were obtained by using the forward selection algorithm to choose the effects and by applying filter 3 to the data. The value of the AR-S indicator, with an accuracy of 0.006, is the same for both of the analysed batteries, achieving a better result than the other presented methods.

Table 5 shows the effects introduced in the consecutive steps of the forward selection algorithm for both of the tested batteries and the value of the AR-S index resulting from the introduction of these additional effects to the model. It should be noted that the first four effects introduced (marked in bold) are the same for both batteries. The model based on these four effects achieves a value of AR-S above 0.95 for each battery. The AR-S value of 0.95 achieved for the first four effects is insignificantly less accurate than the forward selection AR-S value of 0.9629 and 0.9694 for the first and second battery, respectively, when including all of the presented effects, while not overcomplicating the model. From a practical point of view, in our opinion, it can be assumed that the prediction model for both batteries is identical, and thus the two-year exploitation period of the battery had no or very little effect on the performance of the battery itself. Based on this information, the authors propose to use only these four effects (presented in bold in Table 5) for the formulation of the model.

Table 6 shows the values of the coefficients for each of the first four effects indicated in Table 5. The obtained effects clearly define the functional form of the obtained model, which contains the same effects and only slightly differs in coefficients for the tested batteries. The obtained models clearly show that the independent variables of Turns, Distance, SoC and the interaction of SoC * Distance have the greatest impact on the robot’s battery consumption, which was initially considered to be the dependent variable.

Figure 4 shows the distribution of residuals for the designated models of each of the tested batteries. The analysis of the graph clearly shows that the distribution of residuals has the nature of a normal (Gaussian) distribution. Additionally, for approximately 50% of the observations, the value of the predicted energy consumption (EnergyCons) does not differ from the actual battery energy consumption by more than 0.1%. Residuals between the predicted and the actual energy consumption greater than 0.5% do not account for a fraction of more than 2% of all of the observations.

Figure 5 shows the outlier and leverage diagnostics of each of the tested batteries. The visualisation shows the influential observations and how far away the independent variable values of observation are from those of the other observations. The residuals for selected effects presented in Table 5 are shown in Figure 6 and Figure 7 for the tested battery from 2018 and from 2022, respectively. These differences between any data points and the regression model are presented separately for all of the selected effects.

## 4. Discussion

All of the research goals set out in this paper have been met, and the results are as follows:**R1:** Having specified the potential parameters influencing the battery consumption in a single mission and using the data collected from multiple battery discharge cycles under various conditions, a forward selection algorithm was used to find the optimal parameters to formulate the model of battery discharge. After the analysis, it is clear that the parameters with the highest impact on the battery discharge are the current SoC and the level of complexity of the mission itself, which can be described by the distance to travel and the number of turns.**R2:** Having analysed different modelling algorithms combined with different data filters, an optimal modelling algorithm—forward selection algorithm—was chosen. Upon further analysis of the parameters of the model, the authors have decided to limit the parameters to the first four parameters of the model since they were the same for both of the tested batteries, and including additional parameters would not significantly increase the AR-S value, which had been chosen as the indicator of the model’s performance. The AR-S value for the first four parameters was equal to 0.95 while including more parameters could increase this value up to 0.96 (see Table 5). From a practical point of view, the authors suggest using the simpler model for the predictive model of battery discharge. The independent variables used in the model are the number of turns (Turns), the travel distance (Distance), the current SoC of the battery (SoC) and the interaction of variables SoC * Distance. The factors of these variables for both of the tested batteries can be found in Table 6.**R3:** The experiments conducted for both of the batteries have yielded similar results in terms of the most prominent independent variables as well as their factors influencing the battery discharge (see Table 6), which leads us to a conclusion that, in the case of the two tested batteries, the level of exploitation and the age of the battery does not affect the model, though only two batteries do not provide enough evidence and further, more extended tests should be conducted for more units of batteries to verify this claim. Since the models for both batteries are nearly identical, either one of them can be used for predicting the battery discharge in any mission designed for the MiR100 autonomous mobile robot.

## 5. Summary and Conclusions

After analysing different solutions available on the market and the current state of the scientific research, it is clear that no research or solutions are provided to verify whether all of the assigned missions could be executed in the discharge cycle of a single autonomous mobile robot, which usually leads to suboptimal fleet management when scaling up the in-house transportation. As a solution to this problem, the authors first proposed a model based only on the SoC of the robot to determine the potential charge usage in a specific mission but soon realised that such an approach would be impractical in an industrial environment due to huge amounts of data that would need to be collected for each mission individually. The approach proposed in this paper to model the autonomous mobile robot’s battery based on key mission parameters to predict the potential usage of battery charge in any given mission seems to be much more reasonable and yields results appropriate for industrial applications, where even a single execution of a mission could provide the parameters necessary for the prediction to be made for future instances of the mission.

The model created using this approach provides a functional advantage over the existing solutions in the sense that it can be used to predict the future battery discharge of multiple assigned missions, while the existing solutions analyse only the current SoC and compare it with a threshold value, not allowing for any prediction to be made. These solutions are only available for the robots of the specific manufacturer, while the model proposed in this paper could be used in a universal fleet management system, implementing multiple robots from different manufacturers.

## Figures and Tables

**Figure 1 sensors-22-09861-f001:**
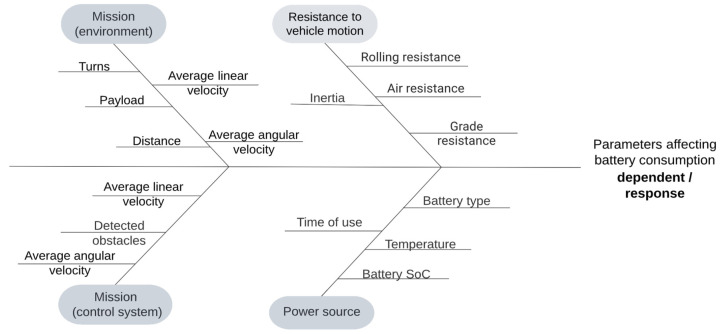
Ishikawa diagram—parameters that may affect energy consumption during mission execution.

**Figure 2 sensors-22-09861-f002:**
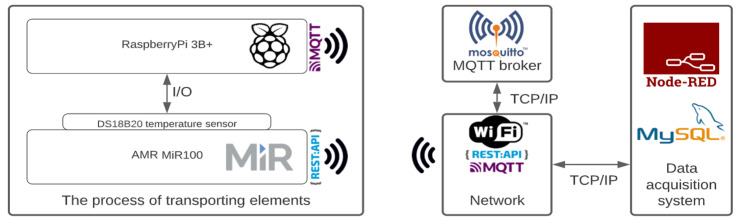
Structure of data acquisition and generation of predictive functions.

**Figure 3 sensors-22-09861-f003:**
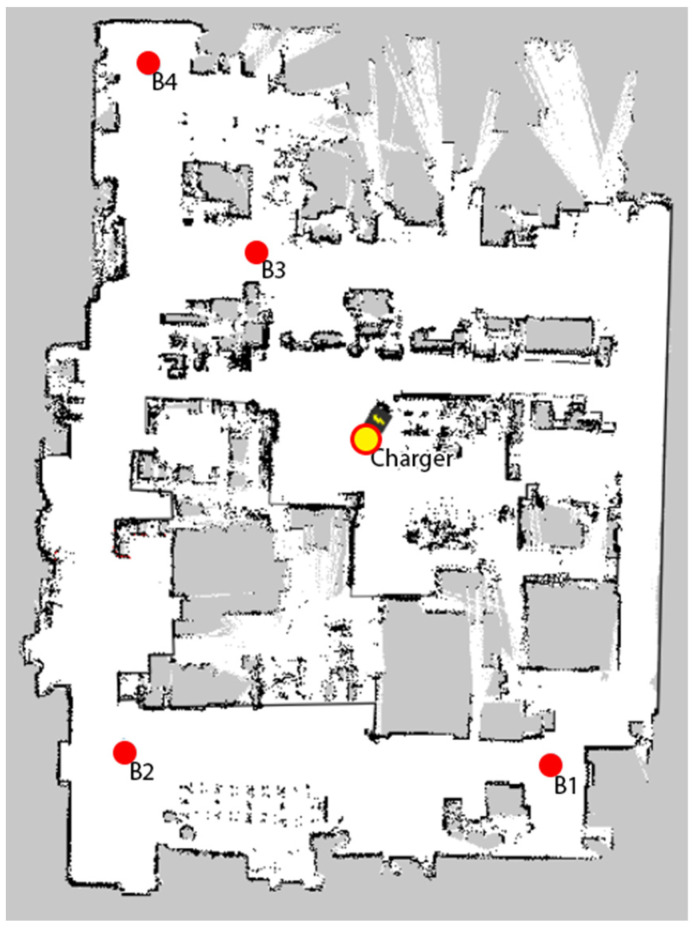
Map of the robot’s working area.

**Figure 4 sensors-22-09861-f004:**
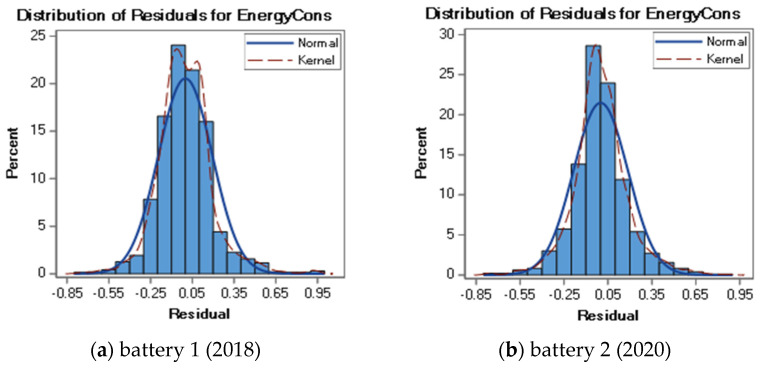
Distribution of residuals for the optimal models of the tested batteries: (**a**) battery from 2018, (**b**) battery from 2020.

**Figure 5 sensors-22-09861-f005:**
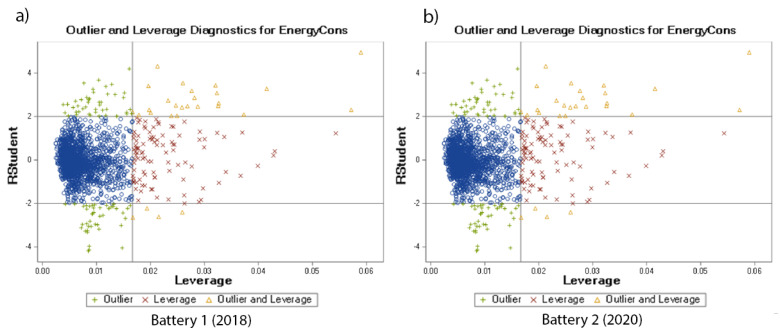
Outlier and Leverage diagnostic for predicted variable of the tested batteries: (**a**) battery from 2018, (**b**) battery from 2020.

**Figure 6 sensors-22-09861-f006:**
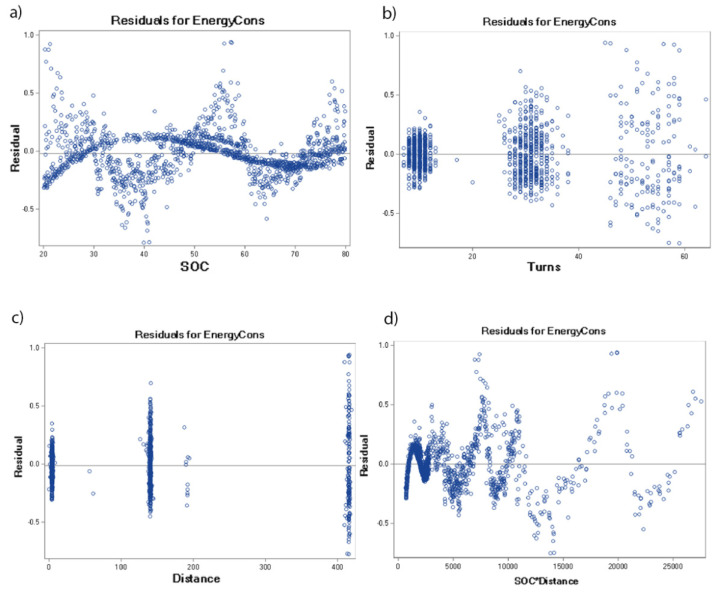
Residuals for selected effects of the tested battery from 2018.

**Figure 7 sensors-22-09861-f007:**
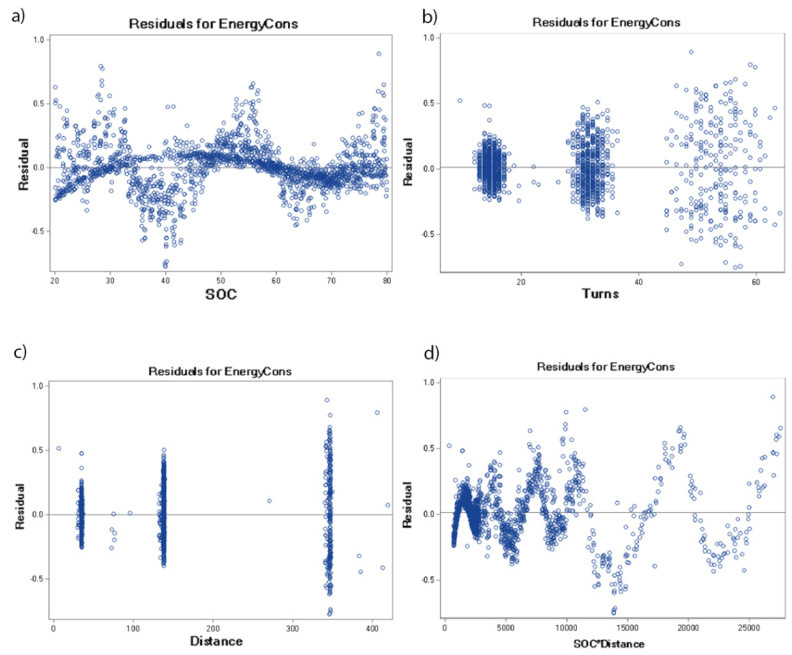
Residuals for selected effects of the tested battery from 2020.

**Table 1 sensors-22-09861-t001:** Parameters to be considered in the experiments.

Measurable Parameters	Irrelevant, Constant or Unpredictable Parameters
Payload Linear velocity Angular velocity Battery SoC Battery Temperature Battery Time of use Number of turns in mission Mission distance	Slope angle—no changes in elevation Rolling resistance and drag coefficient—payload-dependent and mostly unchanged during the execution of the task Battery Type—only Lithium NMC batteries supported for the used robot Detected obstacles—an unpredictable emergency case

**Table 2 sensors-22-09861-t002:** Number of full battery discharge cycles for different mission lengths and payloads.

Payload	Battery 1			Battery 2		
	30 m	140 m	350 m	30 m	140 m	350 m
0 kg	2	2	2	3	4	2
50 kg	2	4	3	3	3	3
100 kg	3	3	3	4	3	4

**Table 3 sensors-22-09861-t003:** Description of variables and their types.

Description of Variable	Label	Type of Variable
Measured energy consumption	EnergyCons	dependent/response
Date of measurement	Date	description/not used
Battery temperature	BatteryTemp.	independent/predictor
Battery SoC	SoC	independent/predictor
Average linear velocity	AvgLinVelocity	independent/predictor
Average angular velocity	AvgAngVelocity	independent/predictor
Number of turns in a mission	Turns	independent/predictor
Payload	Payload	independent/predictor
Average mission travel distance	Distance	independent/predictor

**Table 4 sensors-22-09861-t004:** Experiment results (Adjusted R-Square).

Model Selection	Battery 1				Battery 2			
	No Filter	F1	F2	F3	No Filter	F1	F2	F3
Forward	0.9399	0.9466	0.9568	**0.9629**	0.9489	0.9540	0.9646	**0.9694**
Stepwise	0.9399	0.9467	0.9568	0.9621	0.9489	0.9541	0.9645	**0.9694**
LASSO	0.9355	0.9407	0.9514	0.9523	0.9399	0.9461	0.9540	0.9601
LAR	0.9387	0.9407	0.9527	0.9523	0.9469	0.9487	0.9650	0.9693

**Table 5 sensors-22-09861-t005:** Effects introduced in the consecutive steps of the forward selection algorithm for both of the tested batteries.

Step	Battery 1		Battery 2	
	Effect Entered	Adjusted R-Square	Effect Entered	Adjusted R-Square
**1**	**Turns**	0.9139	**Turns**	0.9167
**2**	**Distance**	0.9196	**Distance**	0.9202
**3**	**SoC**	0.9271	**SoC**	0.9297
**4**	**SoC * Distance**	**0.9508**	**SoC * Distance**	**0.9591**
5	Distance * Distance	0.9521	SoC * SoC	0.9599
6	SoC * SoC	0.9530	BatteryTemp	0.9655
7	AvgAngVelocity	0.9565	Distance * Distance	0.9662
8	SoC * SoC * SoC	0.9573	SoC * Turns	0.9667
9	AvgAngVel * AvgAngVel	0.9613	BatteryTemp * SoC	0.9670
10	SoC * Turns	0.9620	SoC * SoC * SoC	0.9679
11	Payload	0.9624	BatteryTe * BatteryTem	0.9681
12	AvgAngVel * Distance	0.9626	BatteryTemp * Distance	0.9689
13	AvgAng * AvgAng * AvgAng	0.9628	Payload	0.9691
14	SoC * Payload	0.9628	Payload * Payload	0.9692
15	Turns * Payload	0.9629	AvgLinVelocity	0.9693
16			AvgLinVel * AvgLinVel	0.9693
17			Turns * Distance	0.9694

**Table 6 sensors-22-09861-t006:** Selected effects and their factors.

Battery 1		Battery 2	
Effect	Effect Factor	Effect	Effect Factor
Turns	−0.027318	Turns	−0.031285
Distance	0.002327	Distance	0.002999
SoC	−0.141574	SoC	−0.136106
SoC * Distance	0.000000246	SoC * Distance	−0.000037981

## Data Availability

Not applicable.

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
