# Peer review of "Multi-Parameter Predictive Model of Mobile Robot’s Battery Discharge for Intelligent Mission Planning in Multi-Robot Systems"

_sensors, 2022, doi:10.3390/s22249861_

Round 1

Reviewer 1 Report

To improve this paper, I have some comments as follows:

(1)  For introduction section, Literature review should be more detailed and comprehensive. Authors should add more recent research progress to this part and give a brief introduction of the development history on your topic.

(2) It is better for authors to clarify your objectives of your study in the Introduction section.

(3) For your proposed method, advantages compared to other methods should be clarified.

(4) Reasons why you choose this method should be added to your manuscript.

(5) In the comparative study, authors can use other methods to analyze the same problem and highlight your method's accuracy.

(6) For conclusions, more detailed results should be presented.

Author Response

Dear Sir/Madame,

First of all, we would like to thank you very much for your time you spend on reviewing our paper and express our gratitude for your kind review. We would like to thank you for your all remarks and suggestions. They gave us opportunity to improve our paper. In the table below you can find detailed explanations and information about the changes that were made in the body of our manuscript.

Yours faithfully,
Authors

 No.

Remark

Reply

1.

For introduction section, Literature review should be more detailed and comprehensive. Authors should add more recent research progress to this part and give a brief introduction of the development history on your topic.

We have made more detailed review of the recent published papers and made more comprehensive literature. Moreover we gave a brief introduction of the development history in the topic discussed in the paper.

2.

It is better for authors to clarify your objectives of your study in the Introduction section

The objective has been stated in the introduction section as follows:

“The problem therefore comes down to the development of a model determining the future level of battery charge consumption during the operation of a mobile robot, depending on the type of operation being performed, which will allow for the appropriate action to be taken in advance. The purpose of this paper is to determine the factors that have a significant impact on the battery consumption of the MiR100 mobile robot, to develop a predictive model for the battery discharge based on the assigned missions, and to compare the models developed this way for two different batteries from the same manufacturer that differ in the production date and the level of exploitation.”

The research goals have also been mentioned again in the results section of the publication:

“Below are the research goals set out in this publication:

G1. Determination of the parameters that have the greatest impact on the characteristics of energy consumption from Lithium NMC batteries used in the MiR100 autonomous mobile robot.

G2. Finding the optimal model for predicting the consumption of the battery charge level used in the MiR100 autonomous mobile robot.

G3. Comparison of the obtained models for two different batteries from the same manufacturer that differ in the production date and time of use.”

3.

For your proposed method, advantages compared to other methods should be clarified.

An explanation has been added to the related work section:

“The most important advantage of multivariate regression is that it helps understand the correlation between dependent and independent variables. This regression model can also determine the relative influence of one or more predictor variables on the criterion value. The advantage of this model is the ability to identify outliers or anomalies.

The multivariate regression models do not have the disadvantages of other machine learning methods, such as: overfitting (like Decision Tree Regression, Random Forest Regression), poor results on small datasets (like Decision Tree Regression) or compulsory applying feature scaling (like Support Vector Regression).”

4.

Reasons why you choose this method should be added to your manuscript.

The explanation above clarifies the reasons in addition to the reasons previously stated in the existing fragment of the manuscript:

“Despite constituting such a simple assumption about the linear relationship between the outcome and a set of inputs, the regression models proved to work well in many different applications, like medicine [17], industrial process measurements [18], aviation material consumption [19] or traffic flow prediction [20]. The multivariable linear regression models have also been used successfully in battery degradation prediction [21-23].”

5.

In the comparative study, authors can use other methods to analyze the same problem and highlight your method's accuracy.

The purpose of this article was not to compare the regression machine learning methods. The Related Work section describes the practical applications of the selected regression method in battery discharge prediction. Comparative studies with other methods are the reviewer's guideline for further research.

6.

For conclusions, more detailed results should be presented.

In the article, we discussed research goals point-to-point. The definition of research goals is in the Results section. The answers to goals are presented in the Discussion section.

In the revised version of the article, statistical measures such as Leverage diagnostic and residuals are represented in Fig. 5-7.

Reviewer 2 Report

The paper dedicated to development of the universal model of battery discharge based on key parameters. Authors considered a methodology of selecting the key parameters, modelling approach, and algorithms. The model predicts the battery discharge during the given mission based on key parameters. The paper is mostly written well. The authors have provided an overview of the subject in sections Introduction and Related Work, indicating the need of their research. The results obtained are new, interesting, valuable, and mostly well discussed. At the same time, the paper needs some important corrections before its publication. The corrections needed are below. So, the paper needs at least major revision.

Corrections suggested.

1. Please, rewrite Abstract so that it will present clearly the significance of the study and its main results.

2. Please, incorporate section 2 Related work into section 1 Introduction.

3. Lines 171-172.

“The system was based on Node-RED and its architecture is further explained in chapter 3.6.”. As it is not a book but an article, please, use “subsection 3.6” instead of “chapter 3.6”.

4. Lines 207-208.

They should be rewritten as follows:

“where: Fres is the resistance to motion, FA is the air resistance, FG is the gradient resistance, FR is the rolling resistance, FI is the inertia.”.

Similarly, lines 213-215 also should be rewritten.

5. Lines 317-324.

“Below are the research goals set out in this publication: Q1…Q2…Q3…”. In line 317, you wrote about goals but not about questions, so “Q1…Q2…Q3” should be “G1…G2…G3”. Otherwise, please, rewrite sentence in line 317.

6. Section 5 Discussion should be extended providing deeper discussion of the results obtained.

7. Please, capitalize the first letter of words Figure and Table in the text.

8. Please, provide other main parameters for Battery 1 and Battery 2 used.

From my point of view, the actual maximum capacity of a battery (number of its previous recharges) needs also to be added to the model, as a previously heavy used battery with many discharges has a lower maximum capacity at full charge than a new one. So, this feature should also be considered.

9. Line 495.

A sentence “All authors have read and agreed to the published version of the manuscript.” is missing in Author Contributions paragraph.

10. DOIs are not provided for references. Please, provide them when possible.

So, the paper needs major revision.

Author Response

Dear Sir/Madame,

First of all, we would like to thank you very much for your time you spend on reviewing our paper and express our gratitude for your kind review. We would like to thank you for your all remarks and suggestions. They gave us opportunity to improve our paper. In the table below you can find detailed explanations and information about the changes that were made in the body of our manuscript.

Yours faithfully,
Authors

 No.

Remark

Reply

1.

Please, rewrite Abstract so that it will present clearly the significance of the study and its main results.

The abstract has been rewritten to better present the significance of the study.

2.

Please, incorporate section 2 Related work into section 1 Introduction.

The introduction and related work sections have been combined.

3.

Lines 171-172. “The system was based on Node-RED and its architecture is further explained in chapter 3.6.”. As it is not a book but an article, please, use “subsection 3.6” instead of “chapter 3.6”.

Thank you for pointing out the mistake. Chapters have been changed to sections and subsections accordingly.

4.

Lines 207-208. They should be rewritten as follows: “where: F is the resistance to motion, F is the air resistance, F is the gradient resistance, F is the rolling resistance, F is the inertia.”. Similarly, lines 213-215 also should be rewritten.

It has been rewritten as suggested.

5.

Lines 317-324. “Below are the research goals set out in this publication: Q1… Q2…Q3…”. In line 317, you wrote about goals but not about questions, so “Q1…Q2…Q3” should be “G1…G2…G3”. Otherwise, please, rewrite sentence in line 317.

Questions (Q) have been changed to goals (G) in the results section and answers to questions (AQ) have been changed to results (R) in the discussion section.

6.

Section 5 Discussion should be extended providing deeper discussion of the results obtained.

In the article, we discussed research goals point-to-point. The definition of research goals is in the Results section. The answers to goals are presented in the Discussion section.

In the revised version of the article, statistical measures such as Leverage diagnostic and residuals are represented in Fig. 5-7.

7.

Please, capitalize the first letter of words Figure and Table in the text.

It has been corrected as suggested.

8.

Please, provide other main parameters for Battery 1 and Battery 2 used. From my point of view, the actual maximum capacity of a battery (number of its previous recharges) needs also to be added to the model, as a previously heavy used battery with many discharges has a lower maximum capacity at full charge than a new one. So, this feature should also be considered.

Thank you for the suggestion, but the data you have requested is not available in the collected data set. Although it will be considered in future research, at this point it can only be assumed, based on the similarity of the models, that the battery exploitation level does not drastically affect the discharge characteristic. Of course it may be an issue for more heavily exploited batteries, which we cannot currently obtain.

9.

Line 495. A sentence “All authors have read and agreed to the published version of the manuscript.” is missing in Author Contributions paragraph.

It has been added to the Author Contributions paragraph.

10.

DOIs are not provided for references. Please, provide them when possible.

We have added the DOI numbers for cited references.

Reviewer 3 Report

As a continuation of previous work, this paper attempts to create a universal model of battery discharge for any mission based on its key parameters. However, The descriptions of paper need major revisions  before it can be published.

1. Challenges to existing research need to be added to the abstract for further illustrating research motivation.

2. How did the multi-Parameter predictivemodel for mobile robot’s battery presented in this paper differ from the conventional equivalent circuit model (10.1016/j.est.2022.105831) and electrochemical model(10.1002/er.7949), authors needs to be further explained in the introduction.

3. The authors only apply Figure 4 to illustrate the effectiveness of the proposed model, and more curves should be added to further illustrate.

4. The conclusion section is too much, please delete it appropriately.

Author Response

Dear Sir/Madame,

First of all, we would like to thank you very much for your time you spend on reviewing our paper and express our gratitude for your kind review. We would like to thank you for your all remarks and suggestions. They gave us opportunity to improve our paper. In the table below you can find detailed explanations and information about the changes that were made in the body of our manuscript.

Yours faithfully,
Authors

 No.

Remark

Answer

1.

Challenges to existing research need to be added to the abstract for further illustrating research motivation.

The research motivation has been presented more clearly in the abstract.

2.

How did the multi-Parameter predictive model for mobile robot’s battery presented in this paper differ from the conventional equivalent circuit model (10.1016/j.est.2022.105831) and electrochemical model(10.1002/er.7949), authors needs to be further explained in the introduction.

The models presented in the cited publications model the battery itself based on its internal parameters. The model presented in this publication presents the way the battery behaves under the conditions of a specific mission, defined by the identified parameters.

So in a sense, the cited publications estimate the SoC, and this publication estimates the change in SoC for a mission based on its parameters. The proposed model is therefore the next step in battery management, where estimations are based on environmental and mission parameters and not only the battery parameters itself (Current, Voltage).

The difference has been further explained in the introduction.

3.

The authors only apply Figure 4 to illustrate the effectiveness of the proposed model, and more curves should be added to further illustrate.

In the revised version of the article, statistical measures such as Leverage diagnostic and residuals are represented in Fig. 5-7.

4.

The conclusion section is too much, please delete it appropriately.

The conclusion section has been shortened.

Reviewer 4 Report

1. Innovation is not prominent enough.

2. The pseudocode needs to be improved.

3. Grammar and spelling need careful revision, for example, gravitational constant and  resustance to motion.

4. It is not appropriate to cite references in the conclusion.

Author Response

Dear Sir/Madame,

First of all, we would like to thank you very much for your time you spend on reviewing our paper and express our gratitude for your kind review. We would like to thank you for your all remarks and suggestions. They gave us opportunity to improve our paper. In the table below you can find detailed explanations and information about the changes that were made in the body of our manuscript.

Yours faithfully,
Authors

 No.

Remark

Answer

1.

Innovation is not prominent enough.

We have added the additional description to emphasize the novelty of the proposed solution.

2.

The pseudocode needs to be improved.

It has not been specified in how the pseudocode should be improved. Inconsistencies in formatting have been corrected.

3.

Grammar and spelling need careful revision, for example, gravitational constant and resustance to motion.

The manuscript has been revised and the pointed out mistakes have been corrected.

4.

It is not appropriate to cite references in the conclusion.

The reference has been removed from the conclusion.

Round 2

Reviewer 1 Report

The paper can be accepted now. 

Author Response

Dear Sir/Madame,

Would like to thank you again for your time you spent on reviewing our paper and express our gratitude for your kind review. We would like to thank you for your all remarks and suggestions. They gave us opportunity to improve our paper. We are glad that you find our paper proper for publication at current form.

Yours faithfully,
Authors

Reviewer 2 Report

Now, the revised version of the paper looks much better than the previous versions. Authors have made the necessary improvements and corrections to the paper. The paper can be punished in its current form. I only advise to make some further small improvements so that the article fully matches the template in accordance with the instructions to authors.

So, the paper can be accepted in its present form.

Author Response

(The authors gave the same response as above.)

Reviewer 3 Report

It  can be accepted.

Author Response

(The authors gave the same response as above.)

Reviewer 4 Report

Authors made no changes to the pseudocode. The pseudocode looks like code. The parameters given in the pseudocode are also not explained, for example, what's the meaning of initial_data[], getStatus(), mission and current_mission?

  •  

Author Response

Dear Sir/Madame,

Would like to thank you again for your time you spent on reviewing our paper and express our gratitude for your kind review. We would like to thank you for your all remarks and suggestions. They gave us opportunity to improve our paper.

Following your suggestion we have complemented the pseudocode presented in the body of the manuscript.

Yours faithfully,
Authors